**Data Availability Statement:** All relevant data is available within the Dryad Repository at DOI: 10.5061/dryad.m0cfxpp0g.

# Assessment of clinical outcomes with immune checkpoint inhibitor therapy in melanoma patients with CDKN2A and TP53 pathogenic mutations

**Thomas T. DeLeon[1], Daniel R. Almquist[1], Benjamin R. Kipp[2], Blake T. Langlais[3], Aaron Mangold[4], Jennifer L. Winters[2], Heidi E. Kosiorek[3], Richard W. Joseph[5], Roxana S. Dronca[5], Matthew S. Block[5], Robert R. McWilliams[5], Lisa A. Kottschade[5], Kandelaria M. Rumilla[2], Jesse S. Voss[2], Mahesh Seetharam[1], Aleksandar Sekulic[4,6], Svetomir N. Markovic[5], Alan H. Bryce[1] ***

**1** Department of Hematology & Oncology, Mayo Clinic Arizona, Scottsdale, Arizona, United States of America, **2** Department of Laboratory Medicine and Pathology, Mayo Clinic Rochester, Rochester, Minnesota, United States of America, **3** Department of Biostatistics, Mayo Clinic Arizona, Scottsdale, Arizona, United States of America, **4** Department of Dermatology, Mayo Clinic Arizona, Scottsdale, Arizona, United States of America, **5** Department of Hematology & Oncology, Mayo Clinic Rochester, Rochester, Minnesota, United States of America, **6** Mayo Clinic Cancer Center, Phoenix, Arizona, United States of America

* Bryce.Alan@Mayo.edu

## Abstract

### Background

*CDKN2A* and *TP53* mutations are recurrent events in melanoma, occurring in 13.3% and 15.1% of cases respectively and are associated with poorer outcomes. It is unclear what effect *CDKN2A* and *TP53* mutations have on the clinical outcomes of patients treated with checkpoint inhibitors.

### Methods

All patients with cutaneous melanoma or melanoma of unknown primary who received checkpoint inhibitor therapy and underwent genomic profiling with the 50-gene Mayo Clinic solid tumor targeted cancer gene panel were included. Patients were stratified according to the presence or absence of mutations in *BRAF*, *NRAS*, *CDKN2A*, and *TP53*. Patients without mutations in any of these genes were termed quadruple wild type (Quad$^{WT}$). Clinical outcomes including median time to progression (TTP), median overall survival (OS), 6-month and 12-month OS, 6-month and 12-month without progression, ORR and disease control rate (DCR) were analyzed according to the mutational status of *CDKN2A*, *TP53* and Quad$^{WT}$.

### Results

A total of 102 patients were included in this study of which 14 had mutations of *CDKN2A* (*CDKN2A*$^{mut}$), 21 had *TP53* mutations (*TP53*$^{mut}$), and 12 were Quad$^{WT}$. *TP53*$^{mut}$, *CDKN2A*$^{mut}$ and Quad$^{WT}$ mutational status did not impact clinical outcomes including median TTP,

**Funding:** The author(s) received no specific funding for this work.

**Competing interests:** The authors have declared that no competing interests exist.

median OS, 6-month and 12-month OS, 6-month and 12-month without progression, ORR and DCR. There was a trend towards improved median TTP and DCR in $CDKN2A^{mut}$ cohort and a trend towards worsened median TTP in the $Quad^{WT}$ cohort.

## Conclusion

Cell cycle regulators such as *TP53* and *CDKN2A* do not appear to significantly alter clinical outcomes when immune checkpoint inhibitors are used.

## Introduction

Activating mutations of *BRAF* and *NRAS* are the most common mutations observed in melanoma. They are present in approximately 51–63% and 26–28% respectively of the molecular mutations in melanomas [1,2]. Accordingly, the prognostic implications of these mutations are well characterized [3]. Both *BRAF* mutations ($BRAF^{mut}$) and *NRAS* mutations ($NRAS^{mut}$) have been shown to be early events that occur in benign and pre-invasive lesions and are not sufficient to induce carcinogenesis [4,5]. Rather, an accumulation of additional pathogenic mutations is required for pre-malignant $BRAF^{mut}$ or $NRAS^{mut}$ lesions to progress to invasive melanoma [4].

Mitogenic driver mutations such as *BRAF* and *NRAS* induce senescence in premalignant disease and require secondary mutations in cell cycle control genes to convert *BRAF* and *NRAS* aberrations into oncogenes [4,6–9]. Loss of function mutations of *CDKN2A* and *TP53* are two significant genomic alterations that allow oncogene driven melanocytes to overcome senescence and evade apoptosis [10,11]. Given the importance of TP53 and *CDKN2A* mutations in the pathogenesis of invasive melanoma it is understandable that both of these mutations are common mutations in melanoma. According to The Cancer Genome Atlas (TCGA) data genomic alterations of *TP53* and *CDKN2A* are found in 15.1% and 13.3% of melanomas respectively [2], and are are frequently co-mutated with *BRAF* and *NRAS* mutations. When *CDKN2A* mutations are present they are found to be co-mutated with *BRAF*, *NRAS* and non-*NRAS*/*BRAF* mutations at rates of 33.3% - 67.4%, 23.9% - 40.7% and 8.7% - 29.9% respectively [2,12]. Similarly *TP53* is co-mutated with *BRAF*, *NRAS* and non-*BRAF*/*NRAS* mutations with frequencies of 33.1% - 73.9%, 17.4% - 35.7% and 8.7% - 32.2% respectively. *CDKN2A* and *TP53* mutations were present together in 5.5% - 8.3% of cases. *BRAF*, *NRAS*, *CDKN2A* and *TP53* mutations were absent in 8.3% - 32.2% of cases.

Historically both *TP53* and *CDKN2A* mutations are associated with a poor prognosis in melanoma patients. Several studies have shown that patients with *TP53* or *CDKN2A* mutations have a shorter expected survival [13–15]. This occurs, at least in part, because *TP53* and *CDKN2A* mutated tumors are more resistant to chemotherapy [13]. Several preclinical studies have demonstrated that *TP53* and *CDKN2A* mutations lead to a loss of normal cell cycle regulation which in turn causes malignant cells to develop chemoresistance [16,17]. This paradigm of poor outcomes and chemoresistance is pervasive and has been demonstrated in multiple other malignancies. This is further supported by the observation that the use of cyclin-dependent kinase (CDK) inhibitors can enhance responsiveness to chemotherapy in tumors with loss of P16$^{INK4a}$ [CDKN2A loss of function mutation] [18,19]. However, it is not clear that this paradigm remains true in melanoma with the era of checkpoint inhibitors.

Neither *BRAF* nor *NRAS* mutations are thought to directly impact the efficacy of immunotherapy; however, previous studies have demonstrated nuances in response rates with

checkpoint inhibitors according to genotypes. Douglas *et al* were the first to report the influence of $NRAS^{mut}$ on immunotherapy outcomes and concluded that individuals harboring $NRAS^{mut}$ had improved response rates, clinical benefit and progression free survival [20]. However, this study included all subtypes of melanoma and also only contained a small cohort of patients who received programmed death-1 (PD-1) inhibitors. Kim *et al* subsequently published a study assessing the effects of *TP53* and non-V600 BRAF mutations ($BRAF^{non-v600}$) on clinical outcomes of cutaneous melanomas [21]. Neither *TP53* nor $BRAF^{non-v600}$ mutations were associated with overall survival (OS) with ipilumumab treatment. There is a paucity of literature discussing the clinical outcomes of patients with *CDKN2A* and *TP53* mutations since the introduction of immune checkpoint inhibitors. Herein we report the effect of *TP53* and *CDKN2A* mutations on the response to immune checkpoint inhibitors, including PD1 inhibitors, in patients with advanced cutaneous melanoma and melanoma of unknown primary.

## Results

### Mutational status and patient characteristics

A total of 207 melanoma patients had genomic profiling using our in house 50 gene panel, of which 102 patients met the inclusion criteria for this analysis (Fig 1). Genomic profiling was performed between March 1, 2014 and October 1, 2016. Clinical data were collected between January 1, 1990 and April 7, 2017. Of the 102 patients evaluated 14 (13.7%) patients were identified to have $CDKN2A^{mut}$, 21 (20.6%) had $TP53^{mut}$, and 12 (11.8%) were $Quad^{WT}$; the genotypes of $CDKN2A^{mut}$, $TP53^{mut}$ and $Quad^{WT}$ patients are displayed in S1 Fig. The patient characteristics for this cohort of patients are summarized in Table 1.

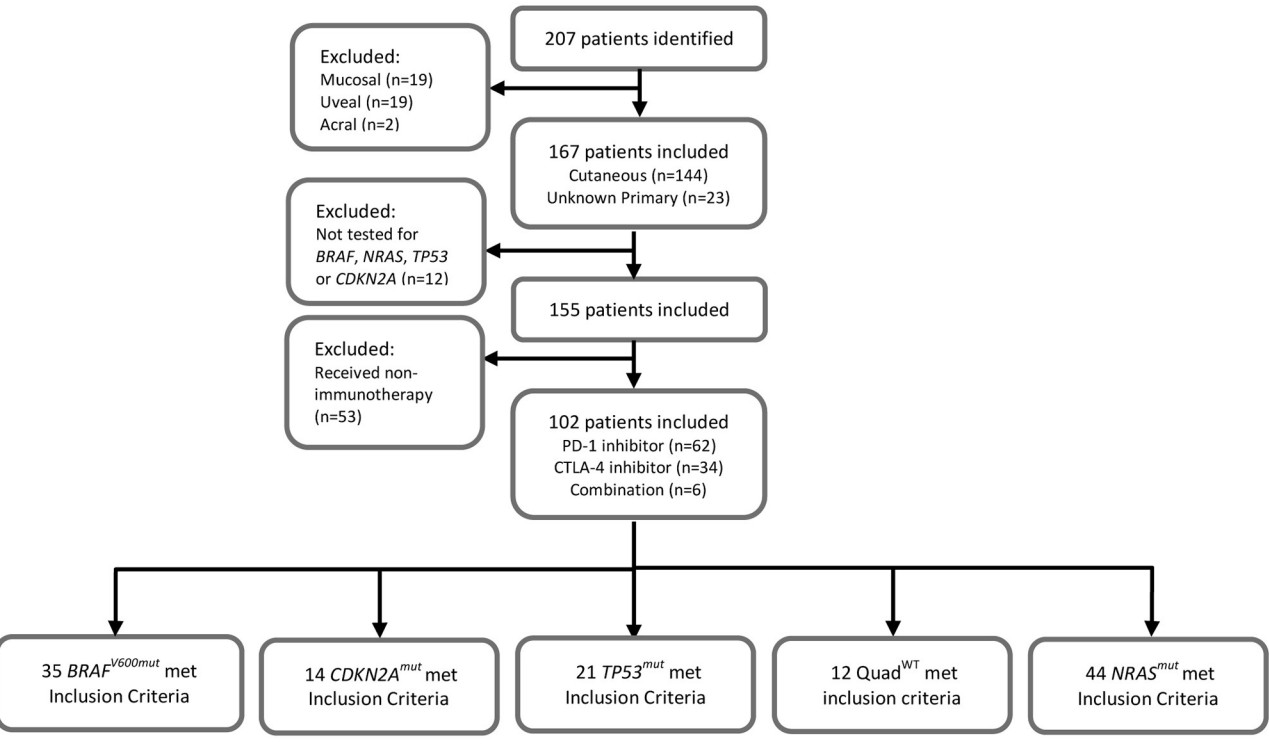

**Fig 1. Flow diagram of patient selection.**

**Table 1. Patient characteristics stratified by TP53 and CDKN2A mutations.**

| | TP53[mut] (N = 21) | TP53[WT] (N = 81) | P Value | CDKN2A[mut] (N = 14) | CDKN2A[WT] (N = 88) | P Value | Quad[WT] (N = 12) | Not Quad WT (N = 90) | P Value |
|---|---|---|---|---|---|---|---|---|---|
| **Age at diagnosis** | | | 0.83 [a] | | | 0.50 [a] | | | 0.15[a] |
| Median | 60.7 | 62.3 | | 63.9 | 61.2 | | 71.9 | 60.8 | |
| Range | (32.4–82.2) | (22.8–91.0) | | (31.6–88.5) | (22.8–91.0) | | (41.3–77.3) | (22.8–91.0) | |
| **Gender** | | | 0.74 [b] | | | 0.87 [b] | | | 0.67[b] |
| Male | 14 (66.7%) | 57 (70.4%) | | 10 (71.4%) | 61 (69.3%) | | 9 (75.0%) | 62 (68.9%) | |
| Female | 7 (33.3%) | 24 (29.6%) | | 4 (28.6%) | 27 (30.7%) | | 3 (25.0%) | 28 (31.1%) | |
| **Ethnicity** | | | 0.05 [b] | | | 0.69 [b] | | | 0.71[b] |
| Caucasian | 20 (95.2%) | 81 (100.0%) | | 14 (100.0%) | 87 (98.9%) | | 12 (100.0%) | 89 (98.9%) | |
| Hispanic | 1 (4.8%) | 0 (0%) | | 0 (0%) | 1 (1.1%) | | 0 (0%) | 1 (1.1%) | |
| **Sites of Disease** | | | | | | | | | |
| CNS | 3 (16.7%) | 20 (26.7%) | 0.38 [b] | 3 (23.1%) | 20 (25.0%) | 0.88 [b] | 4 (36.4%) | 19 (23.2%) | 0.34[b] |
| Liver | 5 (27.8%) | 19 (25.3%) | 0.83 [b] | 5 (38.5%) | 19 (23.8%) | 0.26 [b] | 5 (45.5%) | 19 (23.2%) | 0.11[b] |
| Lung | 10 (55.6%) | 37 (49.3%) | 0.64 [b] | 8 (61.5%) | 39 (48.8%) | 0.39 [b] | 9 (81.8%) | 38 (46.3%) | 0.03[b] |
| Adrenal | 1 (5.6%) | 6 (8.0%) | 0.72 [b] | 2 (15.4%) | 5 (6.3%) | 0.25 [b] | 1 (9.1%) | 6 (7.3%) | 0.83[b] |
| Bone | 4 (22.2%) | 16 (21.3%) | 0.93 [b] | 0 (0%) | 20 (25.0%) | 0.04 [b] | 2 (18.2%) | 18 (22.0%) | 0.78[b] |
| Skin | 2 (11.1%) | 16 (21.3%) | 0.32 [b] | 0 (0%) | 18 (22.5%) | 0.06 [b] | 1 (9.1%) | 17 (20.7%) | 0.36[b] |
| Lymph Node | 8 (44.4%) | 33 (44.0%) | 0.97 [b] | 7 (53.8%) | 34 (42.5%) | 0.44 [b] | 6 (54.5%) | 35 (42.7%) | 0.46[b] |
| Other | 4 (25.0%) | 20 (29.0%) | 0.75 [b] | 6 (50.0%) | 18 (24.7%) | 0.07 [b] | 1 (10.0%) | 23 (30.7%) | 0.17[b] |
| **Melanoma Subtype** | | | 0.10 [b] | | | 0.80 [b] | | | 0.41[b] |
| Cutaneous | 15 (71.4%) | 70 (86.4%) | | 12 (85.7%) | 73 (83.0%) | | 11 (91.7%) | 74 (82.2%) | |
| Unknown Primary | 6 (28.6%) | 11 (13.6%) | | 2 (14.3%) | 15 (17.0%) | | 1 (8.3%) | 16 (17.8%) | |
| **Metastases** | | | 0.32 [b] | | | 0.81 [b] | | | 0.95[b] |
| Yes | 18 (85.7%) | 75 (92.6%) | | 13 (92.9%) | 80 (90.9%) | | 11 (91.7%) | 82 (91.1%) | |
| No | 3 (14.3%) | 6 (7.4%) | | 1 (7.1%) | 8 (9.1%) | | 1 (8.3%) | 8 (8.9%) | |
| **Name of Therapy** | | | 0.14 [b] | | | 0.68 [b] | | | 0.03[b] |
| Pembrolizumab | 7 (33.3%) | 46 (56.8%) | | 8 (57.1%) | 45 (51.1%) | | 7 (58.3%) | 46 (51.1%) | |
| Nivolumab | 1 (4.8%) | 5 (6.2%) | | 0 (0%) | 6 (6.8%) | | 3 (25.0%) | 3 (3.3%) | |
| Ipilimumab | 11 (52.4%) | 23 (28.4%) | | 5 (35.7%) | 29 (33.0%) | | 2 (16.7%) | 32 (35.6%) | |
| Nivolumab/Ipilimumab | 2 (9.5%) | 3 (3.7%) | | 0 (0%) | 5 (5.7%) | | 0 (0%) | 5 (5.6%) | |
| Other therapy | 0 (0%) | 4 (4.9%) | | 1 (7.1%) | 3 (3.4%) | | 0 (0%) | 4 (4.4%) | |
| **Other Therapy Name** | | | - | | | - | | | - |
| Ipilimumab/Dabrafenib | 0 (0%) | 1 (25.0%) | | 0 (0.0%) | 1 (33.3%) | | 0 (0%) | 1 (25.0%) | |
| Ipilimumab/Dacarbazine | 0 (0%) | 1 (25.0%) | | 1 (100.0%) | 0 (0%) | | 0 (0%) | 1 (25.0%) | |
| Pembrolizumab/Indoximod | 0 (0%) | 2 (50.0%) | | 0 (0.0%) | 2 (66.7%) | | 0 (0%) | 2 (50.0%) | |
| **Lines of Therapy** | | | 0.84 [b] | | | 0.43 [b] | | | 0.87[b] |
| 1 | 18 (85.7%) | 68 (84.0%) | | 12 (85.7%) | 74 (84.1%) | | 11 (91.7%) | 75 (83.3%) | |
| 2 | 3 (14.3%) | 10 (12.3%) | | 1 (7.1%) | 12 (13.6%) | | 1 (8.3%) | 12 (13.3%) | |
| 3 | 0 (0%) | 1 (1.2%) | | 0 (0%) | 1 (1.1%) | | 0 (0%) | 1 (1.1%) | |
| 4 | 0 (0%) | 2 (2.5%) | | 1 (7.1%) | 1 (1.1%) | | 0 (0%) | 2 (2.2%) | |
| **LDH elevated [c]** | | | 0.45 [b] | | | 0.06 [b] | | | 0.09[b] |
| Yes | 2 (10.5%) | 16 (21.6%) | | 1 (8.3%) | 17 (21.0%) | | 4 (33.3%) | 14 (17.3%) | |
| No | 13 (68.4%) | 40 (54.1%) | | 5 (41.7%) | 48 (59.3%) | | 8 (66.7%) | 45 (55.6%) | |
| Not tested | 4 (21.1%) | 18 (24.3%) | | 6 (50.0%) | 16 (19.8%) | | 0 (0%) | 22 (27.2%) | |
| **Number of metastatic sites** | | | 0.79 [b] | | | 0.38 [a] | | | 0.35 |
| Median Number of Sites | 1.0 | 2.0 | 0.48[a] | 2.0 | 1.5 | | 2.0 | 2.0 | 0.32[a] |
| **Range** | 0–5.0 | 0–5.0 | | 0–5.0 | 0–5.0 | | 0–4.0 | 0–5.0 | |

(*Continued*)

**Table 1.** (Continued)

| | TP53$^{mut}$ (N = 21) | TP53$^{WT}$ (N = 81) | *P* Value | CDKN2A$^{mut}$ (N = 14) | CDKN2A$^{WT}$ (N = 88) | *P* Value | Quad$^{WT}$ (N = 12) | Not Quad WT (N = 90) | P Value |
|---|---|---|---|---|---|---|---|---|---|
| **Response Rate (ORR)** [d] | | | 0.30 [b] | | | 0.54 [b] | | | 0.73[b] |
| ORR | 9 (47.4%) | 23 (34.3%) | | 5 (45.5%) | 27 (36.0%) | | 5 (41.7%) | 27 (36.5%) | |
| **Disease Control Rate (DCR)** [d] | | | 0.58 [b] | | | 0.15 [b] | | | 0.86[b] |
| DCR | 11 (57.9%) | 34 (50.7%) | | 8 (72.7%) | 37 (49.3%) | | 6 (50.0%) | 39 (52.7%) | |
| **Duration of Immunotherapy (months)** [e] | | | 0.87 [a] | | | 0.50 [a] | | | 0.54[a] |
| Median | 2 | 3 | | 4.0 | 3.0 | | 3.0 | 2.0 | |
| Range | 1.0–9.0 | 0–13.0 | | 0–9.0 | 0–13.0 | | 1.0–9.0 | 0–13.0 | |

[a] Wilcoxon rank-sum test;

[b] Chi square test;

[c] 9 subjects missing LDH data;

[d] 16 subjects missing response data;

[e] 24 subjects with incomplete duration data;

Overall response rate (ORR) = complete response + partial response; disease control rate (DCR) = complete response + partial response + stable disease; TP53$^{mut}$: TP53 pathogenic mutation; TP53$^{WT}$: TP53 wild type; CDKN2A$^{mut}$: CDKN2A pathogenic mutation; CDKN2A$^{WT}$: CDKN2A wild type; Quad$^{WT}$: Quadruple wild type; TTP: Time to progression

Of the 102 patients included in the analysis 93 patients had metastatic disease. Metastases were present in 92.9% of *CDKN2A$^{mut}$*, 85.7% of *TP53$^{mut}$* and 91.7% of *Quad$^{WT}$* patients. The presence of these mutations did not affect the sites of disease with exception of a lack of bone metastases in *CDKN2A$^{mut}$* patients and increased lung metastases in the *Quad$^{WT}$* cohort. Cutaneous melanoma was by far the most common subtype of melanoma, while melanoma of unknown primary was far less common with the latter comprising 14.3%, 28.6% and 8.3% in *CDKN2A$^{mut}$*, *TP53$^{mut}$* and *Quad$^{WT}$* patients, respectively. In the *CDKN2A$^{mut}$* and *Quad$^{WT}$* cohorts PD-1 inhibitors were the most commonly used agents representing 57.1% and 83.3% of immunotherapies respectively, while in the *TP53$^{mut}$* cohort the most common immunotherapy was ipilumumab (CTLA-4 inhibitor) with 52.4% of patients receiving the CTLA-4 inhibitor. Combination immunotherapies were more commonly used in the *TP53$^{mut}$* cohort as compared to the *CDKN2A$^{mut}$* or *Quad$^{WT}$* cohorts. The demographics of the entire cohort were predominantly Caucasian and male.

## Time to progression outcomes

There were no statistically significant differences in TTP identified between the various mutational cohorts (Fig 2). The median TTP for *CDKN2A$^{mut}$* and *CDKN2A$^{WT}$* were 14.0 months (95% CI: 3.0 months–NE) and 6.0 months (95% CI: 3.0–9.0 months) respectively. The median TTP for *TP53$^{mut}$* and *TP53$^{WT}$* were 8.0 (95% CI: 3.0 months–NE) and 6.0 months (95% CI: 3.0–13.0 months) respectively. Those with *Quad$^{WT}$* had a TTP of 3.5 months (95% CI: 2.0 months–NE) versus 6.0 months (95% CI: 4.0–14.0 months) in those that did not have quadruple wild type. All trends were preserved for TTP in the *CDKN2A*, *TP53* and *Quad$^{WT}$* cohorts at 6 and 12-month intervals (Table 2). The proportion of patients without progression at 12-months for *CDKN2A$^{mut}$* and *CDKN2A$^{WT}$* patients were 60.0% (95% CI: 28.5–81.2%) and 38.3% (95% CI: 27.4–49.0%) respectively. For *TP53$^{mut}$* and *TP53$^{WT}$* the percentage of patients without progression at 12-months were 44.4% (95% CI: 22.5–64.4%) and 40.7% (95% CI:

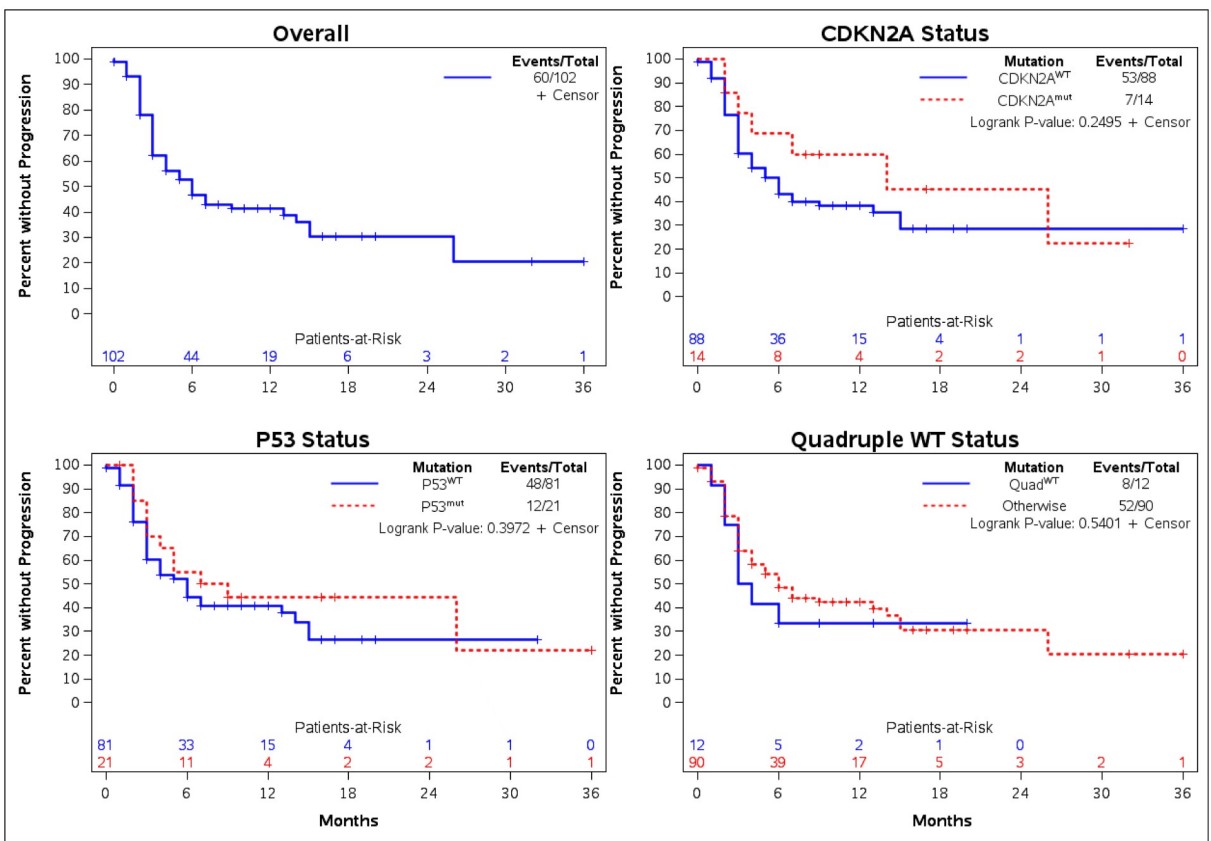

**Fig 2. Time-to-progression by mutation status.**

29.2–51.9%) respectively. The percentage of patients with $Quad^{WT}$ mutational status without progression at 12-months were 33.3% (95% CI: 10.3–58.8%) versus 42.3% (95% CI: 31.2–53.1%) in those without $Quad^{WT}$ status. Fig 2 displays TTP graphs for *CDKN2A*, *TP53* and $Quad^{WT}$ cohorts.

**Table 2. Time to progression by mutation.**

| Mutation | Event/Total | Median Months (95% CI)$^{KM}$ | w/o progression (%) at 6-Months (95% CI)$^{KM}$ | w/o progression (%) at 12-Months (95% CI)$^{KM}$ |
|---|---|---|---|---|
| TP53$^{mut}$ | 12/21 | 8.0 (3.0-NE) | 55.0 (31.3–73.5%) | 44.4 (22.5–64.4%) |
| TP53$^{WT}$ | 48/81 | 6.0 (3.0–13.0) | 44.2 (32.6–55.2%) | 40.7 (29.2–51.9%) |
| CDKN2A$^{mut}$ | 7/14 | 14.0 (3.0-NE) | 68.6 (35.9–87.0%) | 60.0 (28.5–81.2%) |
| CDKN2A$^{WT}$ | 53/88 | 6.0 (3.0–9.0) | 43.2 (32.2–53.7%) | 38.3 (27.4–49.0%) |
| Quad$^{WT}$ | 8/12 | 3.5 (2.0-NE) | 33.3 (10.3–58.8%) | 33.3 (10.3–58.8%) |
| Otherwise | 52/90 | 6.0 (4.0–14.0) | 48.5 (37.3–58.8%) | 42.3 (31.2–53.1%) |
| **Overall TTP:** | **60/102** | **6.0 (4.0–13.0)** | **46.6 (36.2–56.3%)** | **41.2 (30.9–51.2%)** |

CI: Confidence interval; KM: Kaplan-Meier estimate; NE: Not estimable; TP53$^{mut}$: TP53 pathogenic mutation; TP53$^{WT}$: TP53 wild type; CDKN2A$^{mut}$: CDKN2A pathogenic mutation; CDKN2A$^{WT}$: CDKN2A wild type; Quad$^{WT}$: Quadruple wild type; w/o: Without

## Overall survival outcomes

There were no statistically significant differences in OS between the various mutational cohorts (Fig 3). The median OS for $CDKN2A^{mut}$ and $CDKN2A^{WT}$ patients were 41.0 months (95% CI: 17.0–76.0 months) and 57.0 months (95% CI: 26.0 months–NE) respectively. For those with $TP53^{mut}$ and $TP53^{WT}$ mutational status the median OS were NE (95% CI: 21.0 months–NE) and 57.0 months (95% CI: 41.0 months–NE) respectively. The median OS for $Quad^{WT}$ cohort was NE (95% CI: 7.0 months–NE) and those without $Quad^{WT}$ had a median OS of 57.0 months (95% CI: 41.0 months–NE). The proportion of patients alive at 12 months with $CDKN2A^{mut}$ and $CDKN2A^{WT}$ mutational status were 100% (95% CI: 100.0–100.0%) and 74.5% (95% CI: 61.8–83.5%) respectively. The percentage of $TP53^{mut}$ and $TP53^{WT}$ patients alive at 12 months were 87.7% (95% CI: 58.8–96.8%) and 75.4% (95% CI: 62.2–84.6%) respectively. The proportion of patients with 12-month OS in the $Quad^{WT}$ cohort was 70.1% (95% CI: 32.3–89.5%) versus those without $Quad^{WT}$ was 79.5% (95% CI: 67.6% - 87.4%). The OS outcomes are also shown in Table 3.

## Response to immunotherapy

There was no statistically significant difference in overall response rate (ORR) or disease control rate (DCR) between the different mutational cohorts as shown in Table 1. The ORR for $CDKN2A^{mut}$ and $CDKN2A^{WT}$ patients were 45.5% and 36.0% respectively (p-value = 0.54), while the DCR for $CDKN2A^{mut}$ and $CDKN2A^{WT}$ were 72.7% and 49.3% respectively (p-

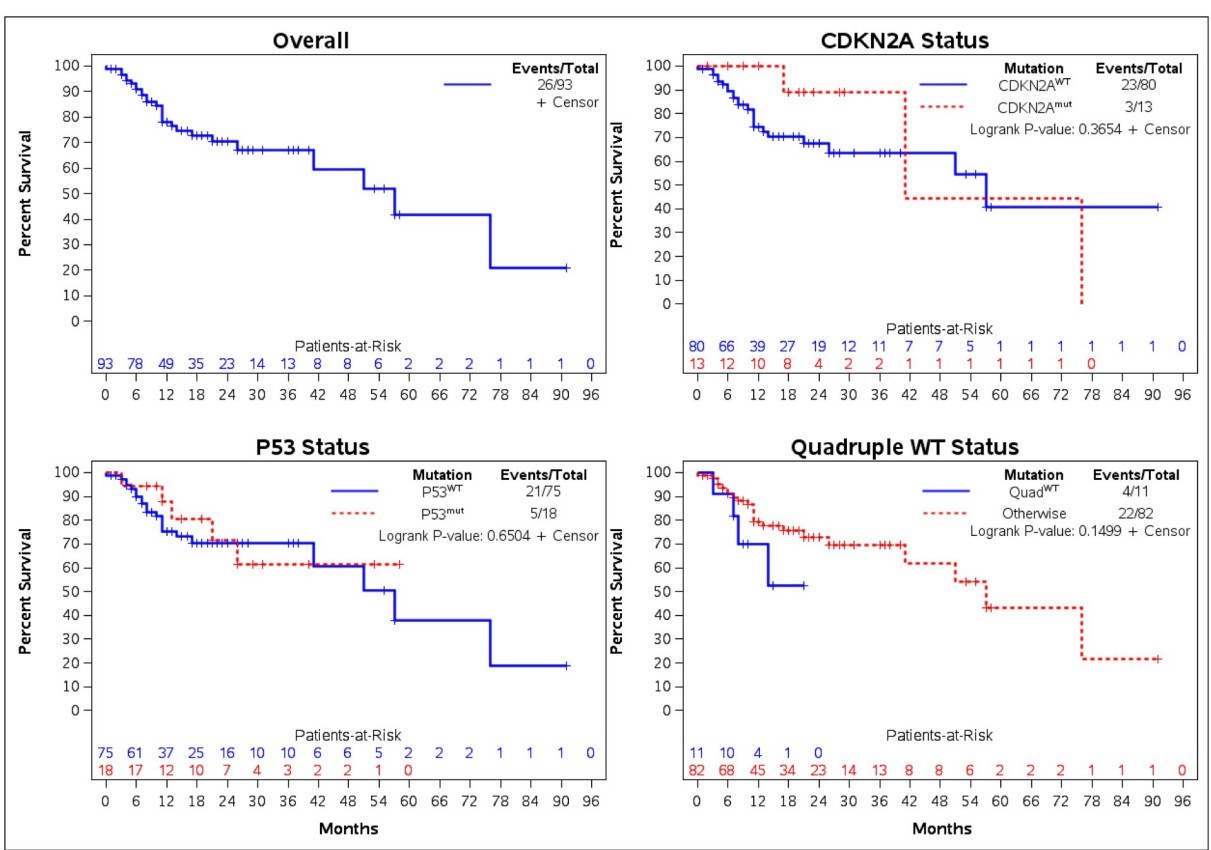

**Fig 3. Overall survival by mutation status.**

Table 3. Overall survival from metastatic diagnosis by mutation.

| Mutation | Event/Total | Median Months (95% CI)KM | Survival (%) at 6-Months (95% CI)KM | Survival (%) at 12-Months (95% CI)KM |
|---|---|---|---|---|
| TP53mut | 5/18 | NE (21.0-NE) | 94.4 (66.6–99.2%) | 87.7 (58.8–96.8%) |
| TP53WT | 21/75 | 57.0 (41.0-NE) | 90.0 (80.2–95.1%) | 75.4 (62.2–84.6%) |
| CDKN2Amut | 3/13 | 41.0 (17.0–76.0) | 100.0 (100.0–100.0%) | 100.0 (100.0–100.0%) |
| CDKN2AWT | 23/80 | 57.0 (26.0-NE) | 89.5 (80.1–94.6%) | 74.5 (61.8–83.5%) |
| QuadWT | 4/11 | NE (7.0-NE) | 90.9 (50.8–98.7%) | 70.1 (32.3–89.5%) |
| Otherwise | 22/82 | 57.0 (41.0-NE) | 90.9 (81.9–95.6%) | 79.5 (67.6–87.4%) |
| **Overall Survival:** | **26/93** | **57.0 (41.0-NE)** | **91.0 (82.7–95.4%)** | **78.2 (67.0–85.9%)** |

CI: Confidence interval; KM: Kaplan-Meier estimate; NE: Not estimable; TP53mut: TP53 pathogenic mutation; TP53WT: TP53 wild type; CDKN2Amut: CDKN2A pathogenic mutation; CDKN2AWT: CDKN2A wild type; QuadWT: Quadruple wild type

value = 0.15). The ORR for $TP53^{mut}$ and $TP53^{WT}$ patients were 47.4% and 34.3% respectively (p-value = 0.30), while the DCR for $TP53^{mut}$ and $TP53^{WT}$ were 57.9% and 50.7% respectively (p-value = 0.58). The ORR for $Quad^{WT}$ and non-$Quad^{WT}$ patients were 41.7% and 36.5% respectively (p-value = 0.73). The DCR for $Quad^{WT}$ patients was 50.0% versus those without quadruple wild type who had a DCR of 52.7% (p-value = 0.86).

## Discussion

Despite the negative prognostic significance typically ascribed to loss of *TP53* in malignancies, the data from this study demonstrates no adverse prognostic or predictive significance for mutations of *TP53* or *CDKN2A* in melanoma patients treated with immune checkpoint inhibitor therapy. The lack of deleterious effect from mutations in genes controlling cell cycle regulators is further supported by consistency across mutational cohorts in regards to TTP, 12-month OS, DCR and ORR. While not statistically significant, $CDKN2A^{mut}$ patients appeared to have a trend towards improved DCR and TTP. However, a larger cohort would be needed to investigate whether clinical outcomes are truly enhanced in patients with $CDKN2A^{mut}$.

The devaluation of cell cycle regulators with immune checkpoint inhibitor therapy is likely explained by the mechanism in which cytotoxic T cells induce cell death. Chemotherapy primarily induces apoptosis in malignant cells via cellular stress and the intrinsic caspase pathway. For instance, many chemotherapy treatments will induce DNA damage, which will in turn signal cell cycle regulators such as *TP53* and *CDKN2A* to activate the intrinsic caspase pathway to induce apoptosis [22–24]. The lethality of immune checkpoint inhibitors is derived primarily from the activation of cytotoxic T cells, which induce apoptosis through granzyme [25]. Granzyme is a serine protease that enters the cytoplasm via perforin and directly activates the caspase pathway independent of cell cycle regulators and induces apoptosis. Additionally, activation of the adaptive immune system will also initiate the extrinsic caspase pathway via death ligands such as tumor necrosis factor (TNF) super family and FasL. Therefore, the cytotoxic effects activated by the adaptive immune system do not appear to be driven by the internal machinery of the cell cycle and its regulators. Rather, T cell recognition of tumor cells via tumor epitopes and immune activating markers that initiates the introduction of granzyme are the more relevant drivers for checkpoint inhibitors. The clinical findings from this small retrospective study support the preclinical rationale that immune checkpoints are not adversely affected by the absence of cell cycle regulators. In addition, these findings are further supported by a previous study that did not show an adverse impact of TP53 mutations on clinical outcomes when patients were treated with ipilimumab [21].

It is more difficult to interpret the findings of the $Quad^{WT}$ cohort given that this cohort includes a diverse collection of mutations. A number of pathogenic mutations were identified in the $Quad^{WT}$ group including: $KIT$ (n = 3), $BRAF^{non-V600}$ (n = 2), $APC$ (n = 1), $CTNNB1$ (n = 1), $HRAS$ (n = 1) and $STK11$ (n = 1). Similar to the $CDKN2A^{mut}$ and $TP53^{mut}$ cohorts there was no statistically significant difference in clinical outcomes observed in this study. However, there was a trend towards worsened median TTP in this patient cohort (3.5 months vs 6.0 months). However, other clinical outcomes including ORR, DCR and 12-month OS were similar between $Quad^{WT}$ cohort and the non-$Quad^{WT}$ cohorts. Given the small size of this cohort (n = 12) and heterogeneous genotype of this cohort conclusions cannot be drawn.

The small size of the study cohort and the retrospective nature of this study are limitations of this exploratory study. Because of the limited sample size the effect of co-mutations on clinical outcomes could not be analyzed with this cohort of patients. Additionally there was heterogeneous use of PD-1 inhibitors and CTLA-4 inhibitors between mutational cohorts. For instance, the $TP53^{mut}$ cohort and non-$Quad^{WT}$ cohort both had a higher proportion of ipilimumab use. Given that PD-1 inhibitors are known to have higher response rates and improved clinical outcomes compared to CTLA-4 inhibitors this may have underestimated the benefit of checkpoint inhibitors in these cohorts. However, despite the difference in treatment modalities there appeared to be consistency across clinical outcomes with similar OS, TTP, ORR and DCR results. Additionally, the $TP53^{mut}$ and non-$Quad^{WT}$ cohorts did not appear to fare any worse despite the higher utilization of ipilimumab.

This exploratory study suggests that immune checkpoint inhibitors are able to function at least as well in the presence of $CDKN2A$ or $TP53$ pathogenic mutations. The lack of clear driver mutations such as $CDKN2A$, $TP53$, $NRAS$ or $BRAF^{V600}$ mutations ($Quad^{WT}$) also did not seem to significantly impact clinical outcomes in this cohort of patients. While the role for $CDKN2A$ and $TP53$ are integral to oncogenesis of melanoma and escape from senescence these mutations do not appear to have a significant deleterious effect on prognosis when immune checkpoint inhibitors are used for therapy.

Given the increasing frequency with which large gene mutation panels are being ordered by practicing clinicians, it is necessary to analyze the significance of common mutations in a given cancer type in order to both focus the clinician on relevant findings, and help them ignore irrelevant ones. Researchers with access to large databases of clinical and genomic findings should systematically analyze the association between common genetic events and clinical outcomes. As in this case, such retrospective studies are exploratory and can help guide larger prospective studies.

## Methods

### Study population/study design

This is a retrospective study which was approved by Mayo Clinic IRB(16–005168). No consent was needed as information was obtain anonymously. This study was conducted in accordance with principles for human experimentation as defined in the Declaration of Helsinki and International Conference on Harmonization Good Clinical Practice guidelines. No participating physicians have conflicts of interest to declare. The Mayo Clinic IRB waived the requirement for informed consent since the data was analyzed anonymously. Patients were identified from all three Mayo Clinic campuses (Minnesota, Arizona and Florida). Patients with a diagnosis of metastatic or unresectable cutaneous melanoma or melanoma of unknown primary whose tumors were analyzed with our 50 gene Solid Tumor Targeted Cancer Gene Panel were included. Patients who received an immune checkpoint inhibitor at any point during their treatment course were included. However, data associated with the first immunotherapy

regimen and overall patient outcomes were evaluated for this analysis. Response to targeted therapy, chemotherapy and subsequent lines of immunotherapy treatments were collected but are not the focus of this study. This study allowed for treatment with cytotoxic T-lymphocyte associated protein 4 (CTLA-4) inhibitors, PD-1 inhibitors, or combinations that included either a PD-1 inhibitor or CTLA-4 inhibitor.

The objective of this study is to investigate the impact of the presence of *CDKN2A* mutations (*CDKN2A$^{mut}$*), *TP53* mutations (*TP53$^{mut}$*) and quadruple wild type (*Quad$^{WT}$*) mutational status on clinical outcomes in patients who received immune checkpoint inhibitors. Patients who did not carry *TP53, CDKN2A, NRAS* or *BRAF$^{v600}$* mutations were termed *Quad$^{WT}$*. The primary endpoint measured was median time-to-progression (TTP) with secondary endpoints including the percentage of participants without progression at 6 and 12 months, median overall survival (OS), OS at 6 and 12 months, disease control rate (DCR) and overall response rate (ORR) to immunotherapy. Response rates were assessed using available CT or MRI imaging and their associated reports. Calculations were based on the best overall response using the immune related response criteria (irRC) and were categorized as complete response (CR), partial response (PR), stable disease (SD) and progressive disease (PD). Pathologic tumor characteristics, patient demographic and clinical details were also collected by chart review.

## Genomic profiling

The Solid Tumor Targeted Cancer Gene Panel is a 50 gene panel that evaluated the following genes: *ABL1, AKT1, ALK, APC, ATM, BRAF, CDH11, CKDN2A, CSF1R, CTNNB1, EGFR, ERBB2, ERBB4, EZH2, FBXW7, FGFR1, FGFR2, FGFR3, FLT3, GNA11, GNAQ, GNAS, HNF1A, HRAS, IDH1, IDH2, JAK2, JAK3, KDR, KIT, KRAS, MET, MLH1, MPL, NOTCH1, NPM1, NRAS, PDGFRA, PIK3CA, PTEN, PTPN11, RB1, RET, SMAD4, SMARCB1, SMO, SRC, STK11, TP53* and *VHL*. This is a laboratory-developed test using Research Use Only reagents. Extracted DNA from the clinical specimen is fragmented, adapter ligated, and a sequence library of fragments is prepared using a custom capture hybridization method. Individual patient samples are indexed for identification and the library is sequenced on an Illumina platform. Sequence data are processed through the Mayo Clinic Clinical Genome Sequencing Lab bioinformatics pipeline and a variant call file is generated for final analysis and reporting (Unpublished Mayo method). This testing is clinically available through Mayo Clinic.

## Statistics

Patient characteristics were compared between mutation statuses (*TP53$^{mut}$* versus *TP53* wild type [*TP53$^{WT}$*], *CDKN2A$^{mut}$* versus CDKN2A wild type [*CDKN2A$^{WT}$*] and *Quad$^{WT}$* versus non-*Quad$^{WT}$*). Wilcoxon rank-sum compared non-normally distributed continuous data and chi-square tests compared categorical data. Nonparametric survival analysis was used to model TTP and OS. TTP was defined as the time from first line immunotherapy date until date of progression. A patient's progression time was censored if they received subsequent treatment, were lost to follow-up, or death occurred before known progression. OS was defined as the time from metastatic diagnosis date until date of death. Survival time was censored when patients were lost to follow-up. Kaplan-Meier (KM) method was used to estimate event rates, median time and 95% confidence intervals. Median TTP and OS estimates were not estimable (NE) where rates were greater than 50% at the last time point in the cohort. Log-rank test was used to compare TTP and OS event rates between mutation statuses. *P* values $\leq$ .05 were considered statistically significant. Analyses were performed in SAS Statistical Software 9.4 (SAS Institute, Cary, NC).

## Supporting information

**S1 Fig. Genotypes *CDKN2A*, *TP53* and quadruple wild type cohorts.**
(DOCX)

## Author Contributions

**Conceptualization:** Aaron Mangold.

**Data curation:** Benjamin R. Kipp, Blake T. Langlais, Aaron Mangold, Jennifer L. Winters, Heidi E. Kosiorek, Richard W. Joseph, Roxana S. Dronca, Matthew S. Block, Kandelaria M. Rumilla, Jesse S. Voss, Mahesh Seetharam, Aleksandar Sekulic, Svetomir N. Markovic, Alan H. Bryce.

**Formal analysis:** Heidi E. Kosiorek.

**Investigation:** Jennifer L. Winters, Kandelaria M. Rumilla, Jesse S. Voss, Mahesh Seetharam, Aleksandar Sekulic, Svetomir N. Markovic, Alan H. Bryce.

**Methodology:** Jennifer L. Winters, Heidi E. Kosiorek.

**Supervision:** Alan H. Bryce.

**Writing – original draft:** Thomas T. DeLeon, Alan H. Bryce.

**Writing – review & editing:** Daniel R. Almquist, Richard W. Joseph, Roxana S. Dronca, Matthew S. Block, Robert R. McWilliams, Lisa A. Kottschade, Kandelaria M. Rumilla, Jesse S. Voss, Mahesh Seetharam, Aleksandar Sekulic, Svetomir N. Markovic, Alan H. Bryce.

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
