## [Decision Letter · Decision Letter 0]

29 Oct 2019

PONE-D-19-27499

Assessment of Clinical Outcomes with Immune Checkpoint Inhibitor Therapy in Melanoma Patients with CDKN2A and TP53 pathogenic mutations

PLOS ONE

Dear Dr. Almquist,

Thank you for submitting your manuscript to PLOS ONE. After careful consideration, we feel that it has merit but does not fully meet PLOS ONE’s publication criteria as it currently stands. Therefore, we invite you to submit a revised version of the manuscript that addresses the points raised during the review process.

In accordance with the expert reviewers, there are only a few minor points to consider. Rather than repeat

those points here, I refer you to the specific remarks (below) for details.

We would appreciate receiving your revised manuscript by Dec 13 2019 11:59PM. To enhance the reproducibility of your results, we recommend that if applicable you deposit your laboratory protocols in protocols.io, where a protocol can be assigned its own identifier (DOI) such that it can be cited independently in the future. For instructions see: http://journals.plos.org/plosone/s/submission-guidelines#loc-laboratory-protocols

We look forward to receiving your revised manuscript.

Kind regards,

Nikolas K. Haass, MD/PhD

Academic Editor

PLOS ONE

Journal Requirements:

2. Please amend the subsection category “[FOR JOURNAL STAFF USE ONLY]” for your manuscript. Unfortunately, this is not a valid category. At this time, please choose one or more subsections that best represent the topic(s) of your study.

Additional Editor Comments (if provided):

see above

Reviewers' comments:

Reviewer's Responses to Questions

**Comments to the Author**

1. Is the manuscript technically sound, and do the data support the conclusions?

Reviewer #1: Yes

Reviewer #2: Yes

2. Has the statistical analysis been performed appropriately and rigorously? 

Reviewer #1: Yes

Reviewer #2: Yes

3. Have the authors made all data underlying the findings in their manuscript fully available?

Reviewer #1: Yes

Reviewer #2: No

4. Is the manuscript presented in an intelligible fashion and written in standard English?

Reviewer #1: Yes

Reviewer #2: Yes

5. Review Comments to the Author

Reviewer #1: DeLeon and colleagues have undertaken a retrospective single centre review of the prognostic and predictive impact of CDKN2A and p53 mutations on TTP/OS and response in immunotherapy treated patients.

The analysis suggests no impact on of these mutations.

I have a number of suggestions which the authors may consider

1) Given the impact of clinicopathological factors on outcome (eg Liver mets, LDH, primary histology) these associations with mutation and outcome should be considered

2) Combining single agent ipilimumab with anti-PD1 based therapy (and even targeted therapy) creates significant heterogeneity. Suggest PD1 based vs ipi alone

Reviewer #2: The manuscript by DeLeon et al addresses an important clinical question as to whether somatic mutations in cell cycle regulators (TP53 and CDKN2A) are predictive of outcomes in melanoma patients treated with checkpoint inhibitor immunotherapy. Although the results show no significant associations, the findings are useful for clinicians in that mutations in either TP53 or CDKN2A need not guide checkpoint inhibitor treatment decisions.

The authors acknowledge that the sample size (n=102 patients) is a major limiting factor that likely influences the study outcomes, however, this will hopefully prompt other melanoma researchers to investigate using larger datasets that will settle the debate.

It would also be interesting in future to delineate between types of checkpoint inhibitors (antiPD-1, antiCTLA-4 or combination) as well as mutation type (likely gain of function vs. loss of function) not possible in this study with low samples sizes for each category.

Minor revisions:

• The word “somatic” mutations should be used in the title and abstract to distinguish from germline mutations which are common in CDKN2A in melanoma.

• When you quote the mutation frequencies as a range it would be clearer to mention you’ve looked at both the Hodis and TCGA datasets.

• “CDKN2A and TP53 mutations were present together” would be better written as “co-occur.”

• “BRAF, NRAS, CDKN2A and TP53 mutations were absent” would be better written as QuadWT (BRAF, NRAS, CDKN2A and TP53) accounted for 8.3%-32.2% of cases.

• “demonstrated in multiple other malignancies” needs a reference.

• “ipilumumab” misspelled and first time appears should mention it targets CTLA-4.

• Consistency in PD1 or PD-1.

• NE should be defined the first time it appears in the text.

• Will the full mutation datasets be made available i.e. the specific single nucleotide variants, is it in the coding or non-coding region? synonymous or non-synonymous? Missense or nonsense etc. How do you know it’s “pathogenic?”

• Why is NF1 not included in the panel? This gene is altered in up to 20% of melanoma cases and BRAF/NRAS/NF1 triple wild-type patients are regarded as a difficult category to treat.

• Why not show your BRAF and NRAS data? You mention in the introduction that the literature is nuanced regarding mutational status vs clinical outcomes for checkpoint inhibitors in melanoma. You have solid numbers in these categories – would your data not help resolve this?

6. PLOS authors have the option to publish the peer review history of their article (what does this mean?). If published, this will include your full peer review and any attached files.

Reviewer #1: Yes: Matteo Carlino

Reviewer #2: Yes: Jessamy Tiffen

---

## [Author Response · Author response to Decision Letter 0]

17 Feb 2020

We thank you and the reviewers for the thorough evaluation of our manuscript and the thoughtful comments. In response to the reviewers’ comments, we are submitting a revised manuscript entitled “Assessment of Clinical Outcomes with Immune Checkpoint Inhibitor Therapy in Melanoma Patients with Somatic CDKN2A and TP53 pathogenic mutations.” We have made appropriate edits to the manuscript, which are highlighted, and address each comment point by point as below in bold. 

Have the authors made all data underlying the findings in their manuscript fully available?

Reviewer #1: Yes

Reviewer #2: No

Response: Once the manuscript has been accepted for publication, we will provide data requested. We will include the data spread sheets and statistics. If there is specific data that is required, I will attempt to obtain but I have included what was used for this project. 

Reviewer #1: DeLeon and colleagues have undertaken a retrospective single centre review of the prognostic and predictive impact of CDKN2A and p53 mutations on TTP/OS and response in immunotherapy treated patients.

The analysis suggests no impact on of these mutations.

I have a number of suggestions which the authors may consider

1) Given the impact of clinicopathological factors on outcome (eg Liver mets, LDH, primary histology) these associations with mutation and outcome should be considered

Response: Thank you very much for your overall favorable impression of our efforts to investigate treatments and outcomes in this rare disease, and for your comments. We agree that these factors do impact the outcomes of patients with this disease. Table 1. represents the patient's baseline characteristics which did include liver mets and LDH. In our data set these clinicopathological factors did not meet statistical significance. They did not play a role in outcomes of our patients. This is likely the result of our study being under powered to determine each factors significance. If we examine a larger population, it would be interesting to see if these play a role. Unfortunately, given the size of our study I'm unable to include more data related to these factors.

2) Combining single agent ipilimumab with anti-PD1 based therapy (and even targeted therapy) creates significant heterogeneity. Suggest PD1 based vs ipi alone

Response: Agreed, adding more agents does create more heterogeneity within treatment paradigms in leads a question of which agent is the most impact full. In our study we attempted to treat patients with standard of care or standard practice within our center. Table 1. Shows that there is no statistical significance and/or difference between the use of these agents in the setting of these mutations. Again, this is likely from our limitation of sample size and power. If this study could be completed on a larger scale there could potentially be a statistical difference identified. At this time given her sample size I'm unable to further expound on this data. 

Reviewer #2: The manuscript by DeLeon et al addresses an important clinical question as to whether somatic mutations in cell cycle regulators (TP53 and CDKN2A) are predictive of outcomes in melanoma patients treated with checkpoint inhibitor immunotherapy. Although the results show no significant associations, the findings are useful for clinicians in that mutations in either TP53 or CDKN2A need not guide checkpoint inhibitor treatment decisions.

The authors acknowledge that the sample size (n=102 patients) is a major limiting factor that likely influences the study outcomes, however, this will hopefully prompt other melanoma researchers to investigate using larger datasets that will settle the debate.

It would also be interesting in future to delineate between types of checkpoint inhibitors (antiPD-1, antiCTLA-4 or combination) as well as mutation type (likely gain of function vs. loss of function) not possible in this study with low samples sizes for each category.

Minor revisions:

• The word “somatic” mutations should be used in the title and abstract to distinguish from germline mutations which are common in CDKN2A in melanoma.

Response: Thank you very much for your overall favorable impression of our efforts to investigate treatments and outcomes in this disease, and for your comments. We agree this is an important term to delineate so we added and addressed this. 

• When you quote the mutation frequencies as a range it would be clearer to mention you’ve looked at both the Hodis and TCGA datasets.

Response: We attempted to make this clear for better understanding for the readers. We edited and addressed this. 

• “CDKN2A and TP53 mutations were present together” would be better written as “co-occur.”

Response: Edited and added. 

• “BRAF, NRAS, CDKN2A and TP53 mutations were absent” would be better written as QuadWT (BRAF, NRAS, CDKN2A and TP53) accounted for 8.3%-32.2% of cases.

Response: Edited and added. 

• “demonstrated in multiple other malignancies” needs a reference.

Response: Added a reference. 

• “ipilumumab” misspelled and first time appears should mention it targets CTLA-4.

Response: Edited and corrected. 

• Consistency in PD1 or PD-1.

Response: Corrected. 

• NE should be defined the first time it appears in the text.

Response: This was a formatting technicality. We have now corrected it. 

• Will the full mutation datasets be made available i.e. the specific single nucleotide variants, is it in the coding or non-coding region? synonymous or non-synonymous? Missense or nonsense etc. How do you know it’s “pathogenic?”

Response: If the publication is accepted we do have variants included in our data sets such as: “TP53:c.722C>T:p.S241F(Ser241Phe):MUT” and “CDKN2A 21971164 A > G (benign)/ CDKN2A 21971186 G > A (PATHOGENIC)/CDKN2A 21971141 C > G ("unknown" in workbench)

CDKN2A 21971164 A > G (benign).” We will include that will inform our readers if the variant is pathogenic according to our NGS. 

• Why is NF1 not included in the panel? This gene is altered in up to 20% of melanoma cases and BRAF/NRAS/NF1 triple wild-type patients are regarded as a difficult category to treat.

Response: This is a wonderful question. We used a 50 gene panel with pre-specified genes which were determined by Mayo Clinic Lab. The panel is not specifically for melanoma but for solid tumors and unfortunately the 50 gene panel did not include NF1. Mayo clinic laboratory created the gene panel and I do not have access to the data on why they chose the specific genes to be included on that panel. It is a wonderful thought and should be evaluated further in a future study. 

• Why not show your BRAF and NRAS data? You mention in the introduction that the literature is nuanced regarding mutational status vs clinical outcomes for checkpoint inhibitors in melanoma. You have solid numbers in these categories – would your data not help resolve this?

Response: This is a challenging question. We wanted to make the focus of our study on CDKN2A and p53 specifically to answer the question if these mutations play a role in response to therapy. We felt that if we included BRAF it would take away from our focus in this manuscript. Our data sets will include the BRAF and NRAS mutations. In a future study looking less specifically at CDKN2A and p53 we may include our data.

---

## [Decision Letter · Decision Letter 1]

27 Feb 2020

Assessment of Clinical Outcomes with Immune Checkpoint Inhibitor Therapy in Melanoma Patients with CDKN2A and TP53 pathogenic mutations

PONE-D-19-27499R1

Dear Dr. Almquist,

We are pleased to inform you that your manuscript has been judged scientifically suitable for publication and will be formally accepted for publication once it complies with all outstanding technical requirements. One important requirement is that the full datasets will need to be deposited to a public repository, in keeping with the PLOS Data policy.

With kind regards,

Nikolas K. Haass, MD/PhD

Academic Editor

PLOS ONE

Additional Editor Comments (optional):

One important requirement is that the full datasets will need to be deposited to a public repository, in keeping with the PLOS Data policy.

Reviewers' comments:

Reviewer's Responses to Questions

**Comments to the Author**

1. If the authors have adequately addressed your comments raised in a previous round of review and you feel that this manuscript is now acceptable for publication, you may indicate that here to bypass the “Comments to the Author” section, enter your conflict of interest statement in the “Confidential to Editor” section, and submit your "Accept" recommendation.

Reviewer #1: All comments have been addressed

Reviewer #2: All comments have been addressed

2. Is the manuscript technically sound, and do the data support the conclusions?

Reviewer #1: Yes

Reviewer #2: Yes

3. Has the statistical analysis been performed appropriately and rigorously? 

Reviewer #1: Yes

Reviewer #2: Yes

4. Have the authors made all data underlying the findings in their manuscript fully available?

Reviewer #1: Yes

Reviewer #2: No

5. Is the manuscript presented in an intelligible fashion and written in standard English?

Reviewer #1: Yes

Reviewer #2: Yes

6. Review Comments to the Author

Reviewer #1: Comfortable with responses to queries raised at initial review ………...…...…………………......……...………....…………...…..

Reviewer #2: My concerns have been adequately addressed however I look forward to seeing full datasets deposited to a public repository upon acceptance, in keeping with the PLOS Data policy.

7. PLOS authors have the option to publish the peer review history of their article (what does this mean?). If published, this will include your full peer review and any attached files.

Reviewer #1: No

Reviewer #2: Yes: Jessamy Tiffen

---

## [Editor Report · Acceptance letter]

9 Mar 2020

PONE-D-19-27499R1 

Assessment of Clinical Outcomes with Immune Checkpoint Inhibitor Therapy in Melanoma Patients with CDKN2A and TP53 pathogenic mutations 

Dear Dr. Almquist:

I am pleased to inform you that your manuscript has been deemed suitable for publication in PLOS ONE. Congratulations! Your manuscript is now with our production department. 

With kind regards,

on behalf of

Prof Nikolas K. Haass 

Academic Editor

PLOS ONE